# Assessment of Welfare in Groups of Horses with Different Management, Environments and Activities by Measuring Cortisol in Horsehair, Using Liquid Chromatography Coupled to Hybrid Orbitrap High-Resolution Mass Spectrometry

**DOI:** 10.3390/ani12141739

**Published:** 2022-07-06

**Authors:** Francesco Cerasoli, Michele Podaliri Vulpiani, Giorgio Saluti, Annamaria Conte, Matteo Ricci, Giovanni Savini, Nicola D’Alterio

**Affiliations:** Istituto Zooprofilattico Sperimentale Abruzzo e Molise, Campo Boario, 64100 Teramo, Italy; f.cerasoli@izs.it (F.C.); g.saluti@izs.it (G.S.); a.conte@izs.it (A.C.); m.ricci@izs.it (M.R.); g.savini@izs.it (G.S.); n.dalterio@izs.it (N.D.)

**Keywords:** cortisol, animal welfare, horse, LC-HRMS/MS

## Abstract

**Simple Summary:**

Identifying distressing practices in horse management can be challenging. There is little research available comparing the stress profile of horses with different management and activities. Currently, there is no full awareness of whether the stabling of the horse is more or less stressful than the free-ranging state. This study evaluated stress in horses with different management and activities by assessing the cortisol levels in the horsehair. All selected horses were from the same area (Central Italy) and in the same conditions according to the European Animal Welfare Indicators (AWIN) protocol. The horsehair was analyzed using liquid chromatography coupled to high-resolution mass spectrometry, which has not been previously used for this purpose. The free-range group of animals displayed much higher levels of cortisol than stabled horses, even higher than in the ones carrying out order service under the Italian State Police; these results demonstrated that horses leaving in the wild can be more subject to stress. Although further studies on this matter are desirable, this research adds an important piece to the stress associated with the management of horses.

**Abstract:**

Horses have always been animals used for companionship, work, transportation, and performance purposes over the history of humanity; there are different ways of managing horses, but studies on how horse welfare is influenced by different activities and managements are scanty. Understanding how the management, the environment, and the different uses of horses can affect the level of stress and well-being is important not only for people associated with horses. Three groups of horses with different management, environments, and activities were selected: (1) stabled horses ridden frequently, (2) horses that perform public order service under the Italian state police, and (3) free-ranging horses. Cortisol analysis was carried out on horsehair samples using liquid chromatography coupled to hybrid orbitrap high-resolution mass spectrometry (LC-HRMS/MS), a laboratory technique used for the first time to quantify horsehair cortisol. The selection of horses to be included in the three groups was carried out by including only subjects with positive welfare assessment in accordance with the horse welfare assessment protocol (AWIN). These analyses demonstrated that the cortisol levels detected in the horsehair of free-ranging animals were significantly higher compared to those detected in stabled and working horses. These results may have been a consequence of complex environmental, managerial, and behavioral factors, which should be worth further investigation

## 1. Introduction

According to the World Organization for Animal Health (OIE) directives, all animals, including horses, should live in good health and an adequate state of well-being. OIE defines animal welfare as “the physical and mental state of an animal in relation to the conditions in which it lives and dies”. An animal experiences good welfare if it is healthy, comfortable, well-nourished, and safe and when it is not suffering from unpleasant states such as pain, fear, and distress, and can express natural behaviors [1]. In 2016, the OIE highlighted important critical issues related to the management of Equidae. An entire chapter of the Terrestrial Animal Health Code has been dedicated to the welfare of Equidae; it applies to horses, donkeys, and mules that are used for or retired from traction, transport, and generation of income. Equids used in sports and competitions, leisure activities, research, or kept solely for the production of meat or biopharmaceuticals, are excluded [2]. 

To increase consumer sensitivity to animal welfare, the European Union has issued a series of welfare directives, which include several categories of animals (calves, hens, laying hens, and pigs) [3,4,5,6,7], but not horses. Neither adequate literature nor comprehensive legislation is available on horse welfare when compared to other farm animals. To address new legislation, more data are therefore needed on the welfare of horses, preferably under different husbandry conditions.

There is an urgent need to have more data on horse welfare, particularly considering different farming conditions. In recent years, cortisol has been proven to be a reliable indicator of stress in animals [8,9,10]. It is commonly used as a biomarker of the activation of the hypothalamic–pituitary–adrenal (HPA) axis. In horses, the cortisol level of horsehair has been used as an indicator of a state of chronic stress [11]. Horsehair is a matrix where cortisol can accumulate for a long time, and, therefore, it can reasonably be regarded as an “archive” of stress [11,12]; even though, authors have found that the stability of the cortisol level in hair is influenced by environmental factors [13]. Relatively few studies are available on the use of hair cortisol for assessing the stress in animals in general, and in horses in particular. There are several areas where it could potentially have positive outcomes for the effective, reliable, and rapid monitoring of the actual state of the animals (sport horses, feral horses, and workhorses).

Different techniques are available for the evaluation of cortisol in hairs and horsehair. Immunoassays [11] and the liquid chromatography–tandem mass spectrometry technique (LC-MS/MS), in particular, LC-HRMS/MS, even with their limits (e.g., indirect determination of analytes, cross-reactivity phenomena), have been recommended for their high selectivity and accuracy due to the use of isotopically labeled internal standards.

Animal Based Measures are currently considered by OIE as the most reliable indicators of animal welfare or stress as they often reflect the outcome of resource inputs and management practices. They should always be used in association with resource-based measures when assessing animal welfare [1]. 

This study aimed to assess the welfare of three groups of horses with different environments, activities, and management by evaluating the cortisol level in the horsehair using the liquid chromatography tandem mass spectrometry technique. 

## 2. Materials and Methods

### 2.1. Animals and Protocols

To select suitable horses to be included in the study, the European Animal Welfare Indicators Project (AWIN) welfare assessment protocol for horses [14] was compiled. AWIN protocols were designed to address the development, integration, and dissemination of animal-based welfare indicators with emphasis on pain assessment and pain recognition. In the framework of our study, the aim of using this protocol was to enroll subjects with a good state of welfare based on the AWIN parameters applied to both categories of horses, free-ranging and stabled, included in the study. Informed consent was obtained from all subjects. 

#### 2.1.1. AWIN Protocol Application

The protocol identifies 4 principles (good feeding, good housing, good health, appropriate behavior), and 12 criteria. Each criterion is assessed based on specific welfare indicators. 

For this study, 10 welfare indicators were considered adequate since they could be assessed in all the three management conditions evaluated. These indicators, always evaluated by the same operator, were:Body condition score (Principle: good feeding, criterion: appropriate nutrition)Body/skin lesions (Principle: good health, criterion: absence of injuries)Lameness (Principle: good health, criterion: absence of injuries)Nasal and ocular discharge (Principle: good health, criterion: absence of disease)Stereotypies and abnormal behaviors (Principle: appropriate behavior, criterion: expression of other behaviors)Respiratory rate, presence or absence of sweating, shivering, apathy, sunburn, (Principle: good housing, criterion: thermal comfort)Vulvar or penis discharge (Principle: good health, criterion: absence of disease)Prolapses (Principle: good health, criterion: absence of injuries)Joint swellings (Principle: good health, criterion: absence of injuries)Availability of water (Principle: good feeding, criterion: absence of prolonger thirst)

#### 2.1.2. Horses

A total of 47, 18 females and 29 males, clinically healthy horses were enrolled in the study. All horses included in the study were between 5 and 15 years of age; this range of age was decided as the animal welfare indicators cannot be evaluated in individuals under 5 years of age [14], while in those above 15 years, the risk of having patients with equine pituitary pars intermedia dysfunction (PPID), which could alter the results, is very high [15]. The selected horses were divided into 3 groups according to the following characteristics: Group 1: This group was composed of 4 females and 12 males of different breeds; 9 animals were Murgese, 2 Hungarian, 1 Sella Italiano, 1 Lipizzan, 1 Maremmano, 1 Oldenburger, 1 Dutch breed). They were stabled alone in boxes (4 × 4 m) with access to the paddock every day. They were fed with hay and feed, (ration balanced by a veterinarian) and shod every 45–50 days. Bedding was composed of dust-free wood shavings. No pathologies or pharmacological treatments were recorded in the last 5 months. The horses included in this group carry out training activities and flatwork 3–4 times a week and work for the Italian State Police (Ladispoli)Group 2: This group included 4 females and 12 males of different breeds: 12 Murgese, 2 Sella Italiano, 1 Selle Francais, 1 Hungarian breed. They were stabled alone in boxes (4 × 4 m), with no access to the paddock. They were fed with hay and feed (ration balanced by a veterinarian) and regularly shod every 45–50 days. Bedding was composed of dust-free wood shavings. The horses of this group carried out every day public order activities in service to the Italian State Police (Rome). In addition, when horses do not carry out these services, they are walked dailyGroup 3: This group was composed of 15 free-ranging horses, 10 females and 5 males; grade horses. They have been in the mountains at an altitude of about 600 m above sea level (Anversa degli Abruzzi—AQ) with access to natural food and hay without supplements, while drinking water was in water tanks. The space, in which these animals live, is not delimited by fences, and only natural shelters are present. The geographic area is also inhabited by wild animals such as deer, wolves, and bears. These horses live together and have no direct contact with humans. These animals were gathered in a fenced area, to proceed with the sampling activity; this acute stress factor does not alter cortisol horsehair concentration.

### 2.2. Cortisol Measurement

#### 2.2.1. Sampling

Animals were sampled in the first week of June.

Methods and procedures used in this study followed the guidelines of the Italian law for the care and use of animals [16]. From each animal, horsehairs (mane hair) including hair bulbs were sampled at the mid-neck region and collected in a sterile box. The quantity taken was about 1 g/animal. Sampling was carried out by plucking the horsehair (mane hair) by hand. The sterile boxes containing horsehair samples were kept at room temperature and delivered to the laboratory within 24 h after the sampling. 

#### 2.2.2. Liquid Chromatography Coupled to Hybrid Orbitrap High-Resolution Mass Spectrometry (HPLC-Q-Orbitrap)

The cortisol present in the horsehair was determined by using liquid chromatography coupled to hybrid Orbitrap high-resolution mass spectrometry.

Cortisol and Internal Standard (IS) Cortisol-d4 (9, 11, 12, 12-d4) stock solutions (Merck KGaA, Darmstadt, Germany) at 1000 and 100 µg mL^−1^ were dissolved in methanol (MeOH) (Biosolve Chimie, Dieuze, France) and stored at −20 °C for four months. The intermediate ones (10, 1 µg mL^−1^, 100, and 20 ng mL^−1^) were freshly prepared in a mixture of MeOH/water (Biosolve Chimie) 50/50 (*v*/*v*) containing 0.05% formic acid (Biosolve Chimie).

Chromatography was performed on a Thermo Ultimate 3000 High-Performance Liquid Chromatography system (San Jose, CA, USA) using an Acquity Ultra Performance Liquid Chromatography (UPLC) BEH C18 column, (50 × 2.1 mm; 1.7 μm, 130 Å; Waters (Milford, MA, USA), connected to an Acquity guard column (5 × 2.1 mm). HPLC eluent A was an aqueous solution containing 0.1% (*v*/*v*) formic acid and eluent B was MeOH. The gradient was started with 40% eluent B for 1 min, continued with a linear increase to 95% B in 8 min, and maintained in this condition for 3 min. The system came back to 40% B in 1 min and was equilibrated for 3 min for a total run time of 15 min. The column compartment and the sample temperature were kept at 30 °C and 16 °C, respectively. The flow rate was 0.25 mL min^−1^ and the injection volume was 10 μL.

A Q Exactive (Q-Orbitrap) mass spectrometer (Thermo Scientific, San Jose, CA, USA) was equipped with heated electrospray ionization (HESI-II) source. The HESI-II and capillary temperatures were set at 320 and 300 °C, respectively, and the electrospray voltage at 3.20 kV (positive ionization mode). Sheath and auxiliary gases were 35 and 15 arbitrary units, respectively. All Q Exactive parameters (resolution, AGC, and IT) were optimized to improve sensitivity and selectivity. MS acquisition was performed using a full scan/dd-MS^2^ experiment. The monitored adduct and product ions such as the collision energy are presented in Table 1.

#### 2.2.3. Sample Preparation

Hair samples were prepared as described by Duran et al. (2017) [11]. Hair was washed three times with 40 µL methanol/mg hair for about 5 min per wash, then dried overnight at room temperature. The first 6 cm of hair next to the root (excluding it) were ground to a fine powder with a TissueLyser II (Qiagen, Hilden, Germany) for 5 min at 30 Hz. After which, 25 mg of samples was weighed in a 2 mL microcentrifuge, plastic tube. Later, 500 microliters of MeOH was added and the extraction was carried out overnight in a shaker. After centrifugation, two more extractions of 5 min duration were performed. The reunited extracts were filtered with a 0.45 µm PVDF (polyvinyliden fluoride) filter, evaporated then re-dissolved in 100 µL of a mixture of MeOH/water 50/50 (*v*/*v*) containing 0.05% formic acid. Finally, the sample was injected into the HPLC-Q-Orbitrap system.

#### 2.2.4. Method Validation

An internal quality control (IQC) was implemented for the analytical batches by adding the IS solution to each sample before extraction (i.e., 12.5 µL of a solution containing the IS a 20 ng mL^−1^). The latter was used with quantitative aims (isotopic dilution). Moreover, a quality control (QC) sample and at least a spiked QC sample at 10 pg mg^−1^ were analyzed to verify the absence of a false negative result. Each real sample was analyzed twice and quantified with a calibration curve in a neat solvent. 

The method was fully validated according to the performance criteria as required by EC Commission Implementing Regulation (EU) 2021/808 and Food and Drug Administration [17,18]. With regard to the quantitative confirmatory methods, the bioanalytical method validation guidance for industry 2018 was followed [18]. Briefly, the analyte was validated at the spiking concentration levels containing 1–100 pg mg^−1^. Good instrumental linearity was observed (R^2^ = 0.9999). Validation data in terms of recovery (trueness) and precision were calculated with the analyte concentrations of spiked samples, obtained from the linear regression equation of calibration standards in solvent (including the IS). Mean recovery was 97%, and repeatability and within laboratory reproducibility were 11% (CV_r, pooled_) and 13% (CV_wR, pooled_), respectively. The limits of detection (LOD) and quantification (LOQ) were estimated based on the recovery and precision observed at the lowest validation level (1 pg mg^−1^). 

#### 2.2.5. Statistical Analysis

Analysis of covariance (ANCOVA) was performed to compare the cortisol level in the three groups using age, sex, and the belonging group as influencing factors. ANCOVA compares a response variable by both factors and continuous independent variables. A general linear model was used to investigate the differences in the mean values of the dependent variable, controlling for the effects of selected other covariates.

## 3. Results

All the animals included in the study were in a good welfare state according to the AWIN protocol parameters [14]. The three groups were compared and no evident differences were highlighted from the application of the AWIN protocol [14], as all the criteria scored “positive” in all groups. All the animals, including the ones kept out-door (free-ranging horses), were found in good health. No lameness and no evident wounds were diagnosed. Stereotypies were neither seen during the visit nor reported by the owners. The body condition score was three out of five, not only in indoor animals, as expected because they all followed the same food regimen as foreseen by Police, but also in free-ranging horses.

Table 2 and Figure 1 report the levels of cortisol detected in the horsehair of the three groups of animals. The values obtained show significant differences between the groups.

Table 3 reports the significance of the ANCOVA concerning cortisol concentration analysis, showing that the three variables (age, sex, and group) accounted for significant variability within the groups. The R^2^ (coefficient of determination), which indicates the % of the variability of the dependent variable explained by the explanatory variables, was 0.51.

Table 4 shows the coefficient and the significance of the explanatory variables considered. The variable that significantly influenced the level of cortisol was the group, while no significant impact of sex and age on the model was derived. If we look at the reported values, for a given age and sex, group 1 and 2 shows lower level of cortisol in comparison with group 3.

## 4. Discussion

The challenges in monitoring horse behavior using a visual survey are the cost, time, and labor intensity [19]. The use of hair cortisol for assessing the stress in horses can noticeably simplify this evaluation, even if the contemporary use of animal and resource-based measures is still necessary for a comprehensive assessment. The cortisol levels found in horsehair of the three groups of animals managed in this study were significantly different. Higher levels were detected in the group of free-ranging horses. Though it is usually assumed that outdoor animals kept under “naturalistic” conditions can experience a better degree of welfare, this assumption indeed depends on the presence of several stress-related variables that have to be considered [20]. Other researchers have found different results; Sauveroche et al., 2020 [21] found no significant differences in horsehair cortisol concentration between the management regimes. According to Korte et al. [22], even if animals are living under natural conditions, they may be exposed to certain constraints (e.g., climatic conditions, feeding problems); these can be considered only as occasional or temporary perturbations, while the expression of natural behaviors is facilitated [23]. Moreover, research has shown that confining horses in individual boxes could reduce opportunities for the expression of natural social behavior with consequent unresponsiveness to the environment [24] and the development of stereotypies [25,26]. Being housed individually within boxes could also induce aggressiveness toward humans [26]. Mazzolaet al. [13] has recently reported that hair cortisol concentration was significantly lower in horses that spent the night indoors, suggesting the best homeostasis for these subjects. This is in agreement with the results of this study where the hair cortisol levels in free-ranging horses were significantly higher compared to those found in police horses. This was observed even though many of these horses were stabled individually. The higher cortisol levels found in free-ranging horses might be due to the fear of being predated during the night; the area where the study was conducted is mountainous with the regular presence of wolves and bears during the whole year. [27] demonstrated that horses that conduct greater activity develop a higher level of plasmatic cortisol, and Hampson [28] has shown how feral horses are able to travel long distances daily; the combination of these two studies could give an explanation for our results, assuming also that our free-ranging horses (group 3) should exercise more than the other two groups. Another study revealed that, on sheep, the evaluation of hair cortisol must have a cautious interpretation because external factors may have a significant influence on the results [29]. The lower cortisol concentration found in the police horses could also be attributed to the fact that these animals had the opportunity to spend part of the day outdoors to be trained and prepared where they have been protected from possible stressful conditions (such as car horns, other animals, people, etc.).

To the best of our knowledge, this is the first study that has used the LC-MS/MS technique for detecting cortisol from horsehair. All the previous studies used immunoassays or ELISA for hair cortisol measurements on horsehair [11,13,21,29,30]. These assays are based on interactions between antigens and antibodies, and are therefore prone to biases due to “similar” molecules, such as other steroids. Conversely, the LC-MS/MS technique is based on physicochemical properties (e.g., molecular formula, mass to charge ratio, and hydrophobicity) so it does not suffer from cross-reactivity phenomena. However, this technique could be affected by matrix effects that can be efficiently reduced by using isotopically labeled ISs, as was performed in this study.

It was observed that the groups formed in our study had an unbalanced sex distribution as there were more females in the free-ranging horses compared to the other two groups. This could have been an important bias in case sex influences cortisol concentration in horsehair, independent of the degree of stress. Literature on this topic is scarce, but in general, research agrees on the lack of sex differences in hair cortisol, particularly in other species such as cows and piglets [31], bears [32], lynx [33], pigs [34], and dogs [35]. Few studies have been carried out on horses; in one case, higher hair concentrations were found in healthy newborn foals compared to healthy adult horses [36]. In another study, adult males showed higher cortisol levels compared to females [37]. In this last case, the higher levels detected in the males were likely due to different stimulation or stressors, as some differences were also highlighted among the females, particularly between mothers with and without offspring. For these mares, the presence of a foal was clearly a stress factor, as they showed regularly higher cortisol levels [37].

It could be noted that in the literature, there is not always a clear correlation between stressors and poor welfare: Campbell [38] claims that the experiencing of temporary negative welfare effects can be important to drive responses, which, in turn, improve welfare through motivating life-sustaining behaviors, and that the absolute eradication of all negative welfare effects is not therefore desirable; differently, Heimbürge et al. showed that a wide array of stressors alter the cortisol concentrations in hair in many species [31].

## 5. Conclusions

An LC-MS/MS method for the determination of cortisol was validated and applied on horsehair for the first time. In this study, even though all enrolled horses reflected clinical and behavioral parameters of good welfare status and good management had been ensured to these animals, it seems that horses stabled individually with access to a paddock recorded a lower level of cortisol compared to those free-ranging. It is noted that group 2 (public order activities in service to the Italian State Police), compared with 1 (training activities and flatwork), had higher values of cortisol. This is likely due to the type of activity carried out (public order service) and to the lack of access to the paddock; these results are in line with the initial expectations. Free-ranging horses included in group 3 (free-ranging) recorded higher levels of cortisol over time even if they could express social behavior, and despite showing, at least apparently, very good health. This study also reveals how good human management of horses, even those subjected to work and training, can provide lower cortisol levels than horses in the wild that live in more natural and less stressful conditions. Further research is also needed in this regard.

## Figures and Tables

**Figure 1 animals-12-01739-f001:**
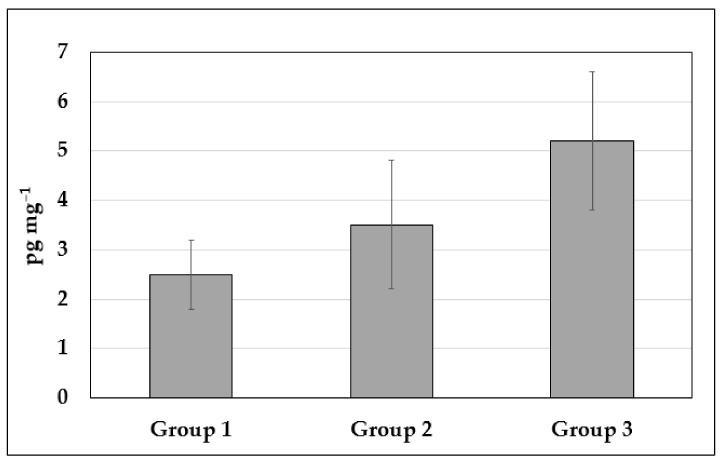
Cortisol concentration in manehair samples belonging to the 3 groups. Results are shown as mean ± standard deviation.

**Table 1 animals-12-01739-t001:** UHPLC-Q-Orbitrap parameters of cortisol and IS.

Analyte	Retention Time (min)	Molecular Formula	Adduct	Monoisotopic Exact Mass (*m*/*z*)	Collision Energy	Fragment 1 ^1^ (*m*/*z*)	Fragment 2 ^1^ (*m*/*z*)
Cortisol	4.5	C_21_H_30_O_5_	[M + Na]^+^	385.1985	30	85.0	288.9
Cortisol-D4	4.5	C_21_H_26_D_4_O_5_	[M + H]^+^	367.2417	-	-	-

^1^ Fragments used only for qualitative purposes (ion ratio calculated in Product Ion scan spectrum).

**Table 2 animals-12-01739-t002:** Detected concentration of cortisol in the three groups—descriptive statistics.

Statistic	Group 1	Group 2	Group 3
**N. of observations**	16	16	15
**Minimum**	**Detected concentration (pg mg^−1^)**		
	1.32	1.36	3.67
**Maximum**	3.64	5.60	8.822
**1st Quartile**	1.931	2.807	4.315
**Median**	2.600	3.449	4.850
**3rd Quartile**	2.832	4.287	5.857
**Mean**	2.475	3.546	5.224
**Variance (n − 1) ***	0.450	1.744	2.027
**Standard deviation (n − 1)**	0.671	1.320	1.424
**Lower bound on mean (95%)**	2.117	2.843	4.435
**Upper bound on mean (95%)**	2.832	4.250	6.012

* Expressed in (pg mg^−1^)^2^.

**Table 3 animals-12-01739-t003:** ANCOVA results and significance.

	DF	Sum of Squares	Mean Squares	F	Pr > F
**Model**	4	60.404	15.101	10.533	<0.0001
**Error**	42	60.099	1.431	-	-
**Corrected** **Total**	**46**	120.504	-	-	-

**Table 4 animals-12-01739-t004:** Model parameters and significance.

	Value	Standard Error	t	Pr > |t|	Lower Bound (95%)	Upper Bound (95%)
**Intercept**	5.216	0.515	10.133	<0.0001	4.177	6.255
**Age**	−0.031	0.055	−0.562	0.577	−0.141	0.079
**Sex-Female**	0.346	0.409	0.847	0.402	−0.479	1.171
**Sex-Male**	0.000	0.000	-	-	-	-
**Group 1**	−2.546	0.485	−5.251	<0.0001	−3.524	−1.567
**Group 2**	−1.476	0.484	−3.051	0.004	−2.452	−0.500
**Group 3**	0.000	0.000	-	-	-	-

## Data Availability

The data generated and analyzed during this study are included in this article.

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
