# Peer review of "Assessment of Welfare in Groups of Horses with Different Management, Environments and Activities by Measuring Cortisol in Horsehair, Using Liquid Chromatography Coupled to Hybrid Orbitrap High-Resolution Mass Spectrometry"

_animals, 2022, doi:10.3390/ani12141739_

Round 1

Reviewer 1 Report

Dear Authors,

your work on horse welfare determined on the basis of observation and on the basis of cortisol content in their hair is very interesting and deserves to be published in Animals. The assumptions of the work are simple and clearly defined, and the research methods are precisely described. The well-described results are backed up by a substantive discussion and the conclusions relate to the results obtained. In my opinion, the conclusions are too extensive, some of the observations can be transferred to the chapter of results or discussion . I understand that the undertaken topic limited the use of a large number of literature items, but it seems to me that the bibliography is still sufficient. In fact, I don't have any serious comments about your work. However, I suggest that in the future, similar research should include horses used in recreation, where, in addition to increased working time, horses are also subjected to the stress of contact with completely new people. It also seems justified to research a group of horses used intensively in sports. Nevertheless, I believe that work brings new values ​​to knowledge about horse welfare.

Author Response

Our reply to the reviewer 1 is reported in the uploaded word file

Reviewer 2 Report

The writing is convoluted and needs to be edited. Please keep it as simple as possible for the reader Eg.

First sentence of Simple Summary is convoluted.  Try:

"Identifying distressing practices in horse management can be challenging.  There is little research available comparing the stress profile of .... "

I have concerns that the science of cortisol in horse hair is not settled.  There seems to be evidence that cortisol levels in hair, although influenced by HPA Axis interactions, can be significantly modified by local impacts (see Salaberger, T., Millard, M., El Makarem, S., Möstl, E., Grünberger, V., Krametter-Frötscher, R., Wittek, T. and Palme, R., 2016. Influence of external factors on hair cortisol concentrations. General and Comparative Endocrinology233, pp.73-78).  This means that horses that are outside all the time in a social group (such as your Group 3) may have increased cortisol stored in hair due to increased parasites, allogrooming or even just local stimulation from rolling in dirt.

You don't seem to address the role of exercise in contributing to increased cortisol secretion see: Gordon et al, 2007.   Please advise.

"aptitudes": I am unclear what this means - not used in the literature.  See Sauveroche, M., Henriksson, J., Theodorsson, E., Holm, A.C.S. and Roth, L.S., 2020. Hair cortisol in horses (Equus caballus) in relation to management regimes, personality, and breed. Journal of Veterinary Behavior37, pp.1-7.

AWIN: Always write out in full anacronyms the first time you use them.

See also above paper for discussion regarding failings of AWIN.  Also "Apathy Principle - does this mean obtunded?

Also Susan V. Horseman, Jo Hockenhull, Henry Buller, Siobhan Mullan, Alistair R.

  1. Barr & Helen R. Whay (2017): Equine Welfare Assessment: Exploration of British Stakeholder

Attitudes Using Focus-Group Discussions, Journal of Applied Animal Welfare Science, DOI:

10.1080/10888705.2017.1283226

The findings are not uncommon - see:

Sauveroche et al, 2020,

Prinsloo, M., Hynd, P., Franklin, S., Weaver, S. and van den Boom, R., 2019. Hair cortisol concentration is inversely related to the severity of equine squamous gastric disease. The Veterinary Journal249, pp.58-59.

Mazzola, S.M., Colombani, C., Pizzamiglio, G., Cannas, S., Palestrini, C., Costa, E.D., Gazzonis, A.L., Bionda, A. and Crepaldi, P., 2021. Do You Think I Am Living Well? A Four-Season Hair Cortisol Analysis on Leisure Horses in Different Housing and Management Conditions. Animals11(7), p.2141.

Please rewrite sentence top of page 2 - horses in developing countries are still working animals, racehorses are working animals.

Please remove the word "anyway" - it does not add anything to the scientific argument.

With respect to the OIE directives - most racing and horse sport organisations are now incorporating the FIVE DOMAINS as a measure of "good"  welfare which go beyond the values discussed in this paper. See: https://ifhaonline.org/resources/IFHA_Minimum_Welfare_Standards.PDF

The Group 3 horses - feral - so contact with humans - even at a distance is stressful. This is not addressed.

In summary, the paper in its present form does little to add to the pre-existing literature.  The authors need to explore the science of hair based cortisol analyses in greater detail and justify why their conclusions within the context of the literature around the interaction between horse keeping and stress.  For example - were any of the horses examined for EGUS? Why wasn't the shave- reshave methodology applied?

Please see Mazzola et al 2021 for similar research.

Please also consider presenting your findings in graphical form.

Author Response

Our reply to reviewer 2 is reported in the attached word file

Reviewer 3 Report

 Must add continuous line numbers

 Title  horse in different environments   not aptitude  also correct abstract

 Were these horses truly feral;. They were managed not  breeding freely. Free-ranging might be a better term   or pasture maintained  Were there fences?

Much more detail on  wild horse environment. Pasture characteristics   Weather. high altitude Any possible source of stress: noise? wolves?

 Abstract horse was not under constant activity. They were ridden frequently

 authors sustained   you probably mean  authors stated or found

2.1.1 shoed should be shod

Need details on bedding and forage for all 3 groups. Time of year of hair sampling should be added

Gp 1 what is level work Do you mean flat work  i.e. no jumping

 Gp 2  moved daily from the box. Are they walked  or something else?

 Gp 3 Much more details needed her  what plants are in the pasture   What about in winter. How many water tanks  and hay sources  in total area?  Any shelter provided?   The horses were with their herd, not alone. Do you mean without human contact?

Put time of sampling in methods not discussion

 Conclusion Worth

 Was mane haired sampled or body hair?  Do you mean 6 cm of hair  ( in which case it has to be mane hair ) or 6 mm?

belonging group   should be group

out-door should be out doors

Discussion 

According to Korte et al. [22], even if living under natural conditions can expose animals to certain constrain  Rewrite  Even if animals are living under natural conditions,  they may be exposed

worth to be deepened with more studies  should be worth further investigation

least stressful

Author Response

Our reply to reviewer 3 is reported in the attached word file

Reviewer 4 Report

This is a potentially interesting paper but at present is subject to a excessive number of fundamental flaws. 

I have taken considerable time to provide extensive feedback to assist the authors with making better use of the data that they have obtained. The text is not helped by a large number of phrasing difficulties and slangy words, which may be as a result of English not being a first language.  Futhermore  the use of several sweeping statements that are not referenced or able to be substantiated must also be avoided.  These occur throughout the manuscript.  

Introduction:  The opening paragraph could be removed as it adds little.  The structure could be improved with greater explanation provided about stress/cortisol. It would be beneficial to discuss stress in general (equine) and how the measurement of cortisol in hair relates to the measurement of cortisol elsewhere.  It is fraught with difficulties leading to some very spurious and misleading interpretations.  The aim needs to be rewritten. 

Method:  at present would not enable replication to be achieved.

Results:  Needs tidying up, simplifying and some of the more method related text removed.  Please do not refer to 'expecting' outcomes, this completely undermines the scientific rigour and therefore casts doubt on the integrity of the work undertaken.    NB only discuss results for which P < 0.05 as signficant; those where P = or > 0.05 are NON signficant and must not be discussed as if they are 'almost signficant'. 

Discussion: some novel text included but no consideration of the multiple other influencing factors is evident. This is a fundamental flaw and casts doubt over the usefulness of the results.  You could refer to similar study findings (does not have to be equids) and how similar findings were dealt with.  At present the discussion content lacks rigour.

Conclusion:  needs to be rewritten - it needs to be shorter and to provide an engaging overview.

Title:  misleading, the word aptitude (which seems to be replaced with 'attitude' in the text) is redundant, there is now assessment or discussion of this concept, whatever it is referring to.   the study seems to be about comparing 'housing conditions' and not aptitude/attitude which I assume relates to 'personality' or something like that?

Summary and abstract:  These are also very confusing, unclear and make statements that are simply untrue.  

Data statement: the raw data are NOT provided in this article.  

I am sorry but at present this submission is not in a form suitable for publication. 

Author Response

Our reply to reviewer 4 is reported in the attached word file

Round 2

Reviewer 2 Report

Please find attached my comments on your rebuttal and redraft.  I strongly urge you to remove all unnecessary filler phrases, adjectives and adverbs from your paper to make it more succinct and easier to read.  You state in your rebuttal that a key reason for doing this research was to validate the LC-MS/MS.  This is not highlighted except as a mention in the introduction.  Most papers I read validating new or different laboratory analysis of biomarkers have a very strict format that is followed (see https://d1wqtxts1xzle7.cloudfront.net/40007382/Comparative_study_of_the_LC-MSMS_and_UPL20151114-19312-7osk23-libre.pdf?1447538386=&response-content-disposition=inline%3B+filename%3DComparative_study_of_the_LC_MS_MS_and_UP.pdf&Expires=1654652658&Signature=Siz8E-APJ8BGjnM4bmRw5qXm~KNGZTtoy7xc4sdHamMCCS0q6cEe0Ik~Fx08O0s05kjAk~I8vErb3j9-HgYTVE1-nygFAKYOrtsdJ1TqV0wWOwCU2xAOfnCs1rcF2VAoSB2nTrsFSpVjbSyCxB4uE2493XnRkeEiOfOeD5dMBXVmb~~iHaUhBaldUmD1Qtd0eVQbO4Y9KacP8zYK7VUnFuVIz-ekqNeikZsdFlg6Ub80oCS1LBz1DudYF-jAfgB-7DZzMEXm621WF22wWb-Npi2ukqIkZGCq9iatVPIlWlwyZ6mA59k4knDtYsfApgEjB71fifjN2cCn3QR8GtnuRQ__&Key-Pair-Id=APKAJLOHF5GGSLRBV4ZA)

Perhaps it would be easier to report the validation process separate to the findings of the different groups of horses re chronic cortisol levels separately.  The paper title should reflect that you are using this research to validate your analytical processes. 

Comments on your rebuttal below: 

Reviewer 2

The writing is convoluted and needs to be edited. Please keep it as simple as possible for the reader Eg.

Thank you for your advice, we have made changes on this purpose

First sentence of Simple Summary is convoluted.  Try:

"Identifying distressing practices in horse management can be challenging.  There is little research available comparing the stress profile of.... "

Thank you for your advice, we have made changes on this purpose

I have concerns that the science of cortisol in horse hair is not settled.  There seems to be evidence that cortisol levels in hair, although influenced by HPA Axis interactions, can be significantly modified by local impacts (see Salaberger, T., Millard, M., El Makarem, S., Möstl, E., Grünberger, V., Krametter-Frötscher, R., Wittek, T. and Palme, R., 2016. Influence of external factors on hair cortisol concentrations. General and Comparative Endocrinology233, pp.73-78).  This means that horses that are outside all the time in a social group (such as your Group 3) may have increased cortisol stored in hair due to increased parasites, allogrooming or even just local stimulation from rolling in dirt.

Really precious observation; we agree with you that the science of cortisol in horsehair is not settled; this is one of the reasons why we have decided to conduct studies in this direction.

NOT addressed.

You don't seem to address the role of exercise in contributing to increased cortisol secretion see: Gordon et al, 2007.   Please advise.

 We think you mean Exercise-induced alterations in plasma concentrations of ghrelin, adiponectin, leptin, glucose, insulin, and cortisol in horses” https://doi.org/10.1016/j.tvjl.2006.01.003 , Gordon et al 2007

in this case Gordon evaluated the plasma concentrations of cortisol but not horsehair cortisol; we agree with you that cortisol plasma concentration can be susceptible to exercise but not in horsehair; it seems to be more reliable in the evaluation of “chronic state” , and that is (cronic evaluation) what we want study

Horses that exercise more will have more cortisol secretion… see Hampson’s work on feral horses in Australia.

"aptitudes": I am unclear what this means not used in the literature.  See Sauveroche, M., Henriksson, J., Theodorsson, E., Holm, A.C.S. and Roth, L.S., 2020. Hair cortisol in horses (Equus caballus) in relation to management regimes, personality, and breed. Journal of Veterinary Behavior37, pp.1-7.

Not cited nor addressed this reference although seems very  relevant. You should include other research that covers the same areas as you have.

Thank you for this precious contribution, we changed the term

AWIN: Always write out in full anacronyms the first time you use them.

Thank you for your advice, we have made changes accordingly

See also above paper for discussion regarding failings of AWIN.  Also "Apathy Principle  does this mean obtunded?  We enriched the description of this point

Also Susan V. Horseman, Jo Hockenhull, Henry Buller, Siobhan Mullan, Alistair R.

1.Barr & Helen R. Whay (2017): Equine Welfare Assessment: Exploration of British Stakeholder

Attitudes Using Focus-Group Discussions, Journal of Applied Animal Welfare Science, DOI:

10.1080/10888705.2017.1283226

Thank you for this observation. We retain that is for sure an important point of view; however the AWIN protocol appear to be one of the most reliable protocols in Europe and especially in Italy concerning horse welfare

You will have an international audience.  You need to at least acknowledge that different parts of the world have different standards.

The findings are not uncommon see:

Sauveroche et al, 2020,

Prinsloo, M., Hynd, P., Franklin, S., Weaver, S. and van den Boom, R., 2019. Hair cortisol concentration is inversely related to the severity of equine squamous gastric disease. The Veterinary Journal249, pp.58-59.

 Not referenced

Tthe paper is ignoring research in the same areas. Please address.

Mazzola, S.M., Colombani, C., Pizzamiglio, G., Cannas, S., Palestrini, C., Costa, E.D., Gazzonis, A.L., Bionda, A. and Crepaldi, P., 2021. Do You Think I Am Living Well? A Four-Season Hair Cortisol Analysis on Leisure Horses in Different Housing and Management Conditions. Animals11(7), p.2141.

Thank you for your observations. For sure this is not the only study considering cortisol in manehair or hair, but as you see the knowledge on this argument is not so settled. Especially there are not many studies that evaluate cortisol values under optimal conditions in different groups of horses.

About the study of Sauveroche et al. 2020 they taken into consideration 25 horses (our study 47) and it is focused on  the evaluation of pathological states; they used ELISA kit during laboratory test unlike our  approach with liquid chromatography tandem mass spectrometry technique (LC-MS/MS).

Our study is mainly aim to demonstrate the differences (cortisol manehair levels) between groups of horses with different management, activities and environment recruiting horses in good condition according with animal-based protocol, and the selection mode is also an innovation.

About the study of Mazzola et al. 2020 they used as laboratory test “Hair cortisol was analyzed using a commercially available assay kit designed to accurately measure the cortisol levels in a variety of sample matrices (Enzo Life Sciences, Farmingdale, NY, USA)” ; as previously mentioned we used different technique wich appears more reliable You need to state WHY it is more reliable… and validate it against previous methods.  I would have thought you would want to get SIMILAR results to prove that your analytical system works.

Please rewrite sentence top of page 2 - horses in developing countries are still working animals, racehorses are working animals.

 NOT ADDRESSED.

Please remove the word "anyway" - it does not add anything to the scientific argument.

 Thank you for your advice, we changed accordingly

With respect to the OIE directives - most racing and horse sport organisations are now incorporating the FIVE DOMAINS as a measure of "good"  welfare which go beyond the values discussed in this paper. See: https://ifhaonline.org/resources/IFHA_Minimum_Welfare_Standards.PDF

 We agree with you but the principal aim of this study is to understand how the cortisol value can change in relation to different environments, activities and management but especially to validate a new reliable laboratory method. WHY is this not captured in the conclusion then?

The Group 3 horses - feral - so contact with humans - even at a distance is stressful. This is not addressed.

We enriched the description of group 3 How were the samples collected for these horses?

In summary, the paper in its present form does little to add to the pre-existing literature.  The authors need to explore the science of hair based cortisol analyses in greater detail and justify why their conclusions within the context of the literature around the interaction between horse keeping and stress.  For example - were any of the horses examined for EGUS? Why wasn't the shave- reshave methodology applied?

Probably we have not highlighted enough that, in our opinion, it is the laboratory technique that has the character of innovation and uniqueness in this study. The final aim could be to use this technique in the future to evaluate well- being combined with the animal based measure assessment. However, the knowledge on the evaluation of well-being through horsehair is not settled (as you mentioned in a previous comment); we retain that improving knowledge on this argument is still a crucial point on animal welfare science.

Then you need to highlight this throughout the study.  The fact you are seeking to validate the analysis side of this experiment does not excuse you from addressing the underlying science. This aim needs to be reflected in the title.

Please see Mazzola et al 2021 for similar research.

Similar reasearch but laboratory technique is different and probably less accurate “ Hair cortisol was analyzed using a commercially available assay kit designed to accurately measure the cortisol levels in a variety of sample matrices….” In our study we used  “liquid chromatography tandem mass spectrometry technique (LC-MS/MS), in particular LC-HRMS/MS have been recommended for their high selectivity and accuracy due to the use of isotopically labeled internal standards….”. This technique appears more reliable, we have also another manuscript (which we will present after this one) to demostrate it.

I understand.  You are seeking to show that your analytical techniques are more accurate so you need to state this in the paper. You have not provided any real evidence that Elizas are less accurate than LCM?

Please also consider presenting your findings in graphical form.

We would prefer to use tables because they include all possible information for each group (variance, median, mean etc.), while it would be difficult to present all of them in graphical form. If we report in a graph only one or two of these value (e.g. mean with SD) we would risk to duplicate data already given in tables. However there are just 3 groups of values, therefore we think that it would be easy, for the reader, to find the main information in the tables. 

I disagree.  Put a graph in for the visually oriented reader and put your full data in as an appendix.

According to Korte et al. [22], even if animals are living under natural conditions, they may be exposed to certain constraints (e.g. climatic conditions, feeding problems), these can be considered only as occasional or temporary perturbations, while the expression of natural behaviors is facilitated [23].

SUGGEST.

Animals are able to express a full range of behaviours when managed under natural conditions.  This may include experiencing periods of deprivation, parasitism, predation and frustration (e.g. competing for mates) which will increase cortisol levels (22?).  There is an inherent inconguity applying the OIE standards of comfortable, well-nourished and safe when experiencing pain, fear and distress is part of expressing natural behaviors.

Surely if you were validating this analysis technique you would want to:

a. validate it against the accepted ELIZA values by testing the same samples with an accepted ELIZA test and looking for correlation

b. find the same results as other researchers have found using ELIZAs?

Reviewer 3 Report

p 2 introduction

 it applies to horses, donkeys, and mules that are destined, used for  Omit destined 

p 4 Bedding composed by wood shavings dust free  Composed of dust-free wood shavings 

p 6    Results foundin should be found in 

p 10 looked clearly as a stress factor, as they showed regularly higher cortisol levels [33].   should be was clearly 

Reviewer 4 Report

Thank you for the changes made in response to all reviewers.  I have a few further comments to make, all intended to improve the manuscript.  

Introduction

Remove parag 1 – wasted words does not add to the content, and can come across as quite condescending to the informed reader, especially one who is an equine science researcher.

On page 2 ‘farming conditions’ should probably be ‘husbandry conditions’ to more accurately reflect how horses are kept by non-livestock producers

Top of page 3 – please rephrase ‘Not many studies are available’ to something like ‘relatively few’ (this then also covers the concerns raised by the other reviewers as well)

Middle of page 3 - ‘This study aim to’ should be ‘This study aimed to’

Section 2.1 is still confusing as to what was done.   subjects recruited to be assessed using AWIN and then when these assessments were done some of these subjects (depending on AWIN scores) were enrolled in this the study?  How were subjects selected on the basis of other factors besides AWIN scores?  As you can see I would not be able to replicate the study because it is not completely clear in the existing/revised text.  (sorry)

A related note if I may, I appreciate that AWIN is used frequently in Italy and some European countries, but Animals has an international readership, including countries that are not so familiar with AWIN, or simply do not use it (they may use other systems).  It is therefore important to explain what it is fully as requested in my original review and noted by other reviewer/s.

Tables (and figures) still need to have full legends, the point is that the table content should be fully understandable if the reader has not read any preceding text, such as the materials and methods. 

2.1.1  How was this ‘same operators’ scores validated?

2.1.2  Please provide the justification of ‘age has been decided as the animal welfare indicators cannot be evaluated in individuals under 5 years of age [15] while in those above 15 years the risk of having patients with PPID disease, that could alter the results, is very high [16]’ – a large proportion of horses in ownership world over would be <5 or 15+. How useful is an indicator that excludes such a large proportion.  (Sorry this may be a question for the AWIN creators, however should probably be noted in your paper.)

2.1.2  Please spell out PPID fully. Please also note that it should be referred to as a disorder not a disease. Also there other conditions e.g. EMS that may be as problematic to the use of AWIN as PPID; this should also be acknowledged alongside text on PPID.   

Page 4 – re. groups.  ‘composed by’ should be ‘composed of’ or even ‘comprised’….

2.2.1  perhaps replace ‘tearing’ with ‘plucking’ – tearing sounds somewhat barbaric and painful?

Re. Table 3 – re. ‘bring significant information to the model’ could be rephrased in terms of accounting for significant variability within xxxx  (just a bit more conventional wording).

Discussion

Much improved - thank you

Parag 1- currently concerning…   suggests that it is Ok to just wait and do an overall analysis of what HAS happened to the individual as evidenced by hair cortisol .. not what is happening at the current time..  This does not set the scene very usefully in terms of why hair cortisol is worth looking at. 

Replace ‘hamper’ with ‘reduce opportunities’ or even ‘prevent’

I don’t think that the statement ‘…in horses that spent the night indoors, suggesting the best homeostasis for these subjects’ is helpful at all, or necessarily correct.  What is meant by homeostasis needs to be explained, and the physical AND mental impacts need to be considered – this is especially important as this paper appears to be keen contribute to the body of contemporary animal(equine) welfare research.

re. ‘police horses that spent most of their life stabled’ – this statement requires a reference please.

Conclusion

Suggest using the descriptors for group 1 and group 2 as some readers may only read conclusion … when deciding whether to read the entire paper.  it also helps emphasise the focus of your findings/evaluation.

(I have not provided an annotated R1 manuscript.)

Round 3

Reviewer 2 Report

I am still struggling with the dichotomy that the authors insist on supporting that low levels of hair cortisol in stabled/domesticated horse reflects good animal welfare and higher levels of cortisol in the hair of the free ranging animals represents poor animal welfare even though they state they are in agreement that the science around cortisol and hair and subsequent judgements around welfare is not settled.  I have asked the authors to review Mellor's work on the five domains and to consider that horses, like all animals, need to experience aversive stimuli in order to learn, adapt and ultimately live a life that enables them to express all the behaviours associated with being a horse.   Like most things in biology, the stress vs welfare picture is not black and white.  That said, the paper should be published with the minor corrections as outlined in the attached file.  Perhaps the nuance in cortisol levels over time will be better defined with this new analytical approach.  That is a good outcome.  Well done.  

Author Response

Please see the attachment, you will find our answers directly going through into your notes.

Kind Regards

Michele Podaliri Vulpiani

Reviewer 4 Report

Thank you for your attention in round 2, the manuscript is much clearer now. 

Author Response

Dear reviewer,

we thank you so much for your revisions, suggestions and patience to helping us to improve the manuscript.